# A new approach to Health Benefits Package design: an application of the *Thanzi La Onse* model in Malawi

**Margherita Molaro**[1]*, **Sakshi Mohan**[2], **Bingling She**[1], **Martin Chalkley**[2], **Tim Colbourn**[3‡], **Joseph H. Collins**[3‡], **Emilia Connolly**[4‡], **Matthew M. Graham**[5‡], **Eva Janoušková**[3‡], **Ines Li Lin**[3‡], **Gerald Manthalu**[6‡], **Emmanuel Mnjowe**[7‡], **Dominic Nkhoma**[7‡], **Pakwanja D. Twea**[6‡], **Andrew N. Phillips**[3‡], **Paul Revill**[2‡], **Asif U. Tamuri**[5‡], **Joseph Mfutso-Bengo**[7], **Tara D. Mangal**[1,2], **Timothy B. Hallett**[1]

1 MRC Centre for Global Infectious Disease Analysis, Jameel Institute, School of Public Health, Imperial College London, London, United Kingdom, 2 Centre for Health Economics, University of York, York, United Kingdom, 3 Institute for Global Health, University College London, London, United Kingdom, 4 London School of Hygiene and Tropical Medicine, London, United Kingdom, 5 Centre for Advanced Research Computing, University College London, London, United Kingdom, 6 Department of Planning and Policy Development, Ministry of Health and Population, Lilongwe, Malawi, 7 Kamuzu University of Health Sciences, Blantyre, Malawi

‡Listed in alphabetical order.
* margherita.molaro@ic.ac.uk

**Data Availability Statement:** The Thanzi La Onse model is open source and available for review and usage at https://github.com/UCL/TLOmodel. In particular, the outputs analysed in this study can be

## Abstract

An efficient allocation of limited resources in low-income settings offers the opportunity to improve population-health outcomes given the available health system capacity. Efforts to achieve this are often framed through the lens of "health benefits packages" (HBPs), which seek to establish which services the public healthcare system should include in its provision. Analytic approaches widely used to weigh evidence in support of different interventions and inform the broader HBP deliberative process however have limitations. In this work, we propose the individual-based *Thanzi La Onse* (TLO) model as a uniquely-tailored tool to assist in the evaluation of Malawi-specific HBPs while addressing these limitations. By mechanistically modelling—and calibrating to extensive, country-specific data—the incidence of disease, health-seeking behaviour, and the capacity of the healthcare system to meet the demand for care under realistic constraints on human resources for health available, we were able to simulate the health gains achievable under a number of plausible HBP strategies for the country. We found that the HBP emerging from a linear constrained optimisation analysis (LCOA) achieved the largest health gain—∼8% reduction in disability adjusted life years (DALYs) between 2023 and 2042 compared to the benchmark scenario—by concentrating resources on high-impact treatments. This HBP however incurred a relative excess in DALYs in the first few years of its implementation. Other feasible approaches to prioritisation were assessed, including service prioritisation based on patient characteristics, rather than service type. Unlike the LCOA-based HBP, this approach achieved consistent health gains relative to the benchmark scenario on a year-to-year basis, and a 5% reduction in DALYs over the whole period, which suggests an approach based upon patient characteristics might prove beneficial in the future.

reproduced from model tag
"Molaro_et_al_2024_HBP_design" (accessible at
https://github.com/UCL/TLOmodel/tags), using the
scenario file src/scripts/healthsystem/
impact_of_policy/scenario_impact_of_policy.py.
All analysis scripts used to generate the plots in the
manuscript are located in the same directory and
have filenames beginning with
"analysis_impact_of_policy_". In addition, post-
processed output data can be directly obtained
from Zenodo at https://doi.org/10.5281/zenodo.
13627185.

**Funding:** This project is funded by The Wellcome
Trust (223120/Z/21/Z to TBH) and contributed to
the salaries of MM, BS, and TM. MM, BS, TM, and
TBH acknowledge funding from the MRC Centre
for Global Infectious Disease Analysis (reference
MR/X020258/1), funded by the UK Medical
Research Council (MRC). This UK funded award is
carried out in the frame of the Global Health
EDCTP3 Joint Undertaking. The funders had no
role in study design, data collection and analysis,
decision to publish, or preparation of the
manuscript.

**Competing interests:** The authors have declared
that no competing interests exist.

## Author summary

All publicly funded healthcare systems face difficult decisions about how limited resources should be allocated to achieve the greatest possible return in health. These decisions are particularly pressing in lower-income countries (LICs) like Malawi, where resources are extremely limited and their inefficient allocation results in larger morbidity and mortality. In this work, we introduce a new analytical tool to inform such decisions based on an "all diseases, whole healthcare system" simulation specifically tailored to Malawi, the *Thanzi La Onse* (TLO) model. The TLO model is able to forecast the health burden that should be expected from different resource-allocation strategies in Malawi specifically, allowing policy-makers to explore a wide range of policy options in a safe and theoretical fashion. In this analysis, we compare the forecasted health burden from a set of common resource-prioritisation strategies, and draw some general conclusions as to what makes certain strategies more or less effective in reducing the health burden incurred.

## 1. Introduction

How should limited resources be allocated to achieve the greatest possible return in health? This difficult question is one every public healthcare provider must grapple with; it is however a particularly urgent one in low-income settings, where available financial resources are limited, and their inefficient allocation results in a much larger loss of health and life [1].

A widely adopted strategy to address this question and inform the allocation of healthcare resources in many low- and middle- income countries is through the design of "health benefits packages" (HBPs). HBPs seek to establish which services the public healthcare system should focus on delivering, given that it is unable to take financial responsibility for the full range of possible services [2].

Although the objectives of HBPs are well defined, focusing typically on efficiency and equity in resource allocation, identifying the appropriate design for HBPs is far more challenging [3,4]. The complex generative processes undertaken by stakeholders must address a wide range of concerns, both quantitative and qualitative in nature, which include (but are not limited to) burden of disease, cost-effectiveness (CE), budget impact, cultural and political acceptability, financial risk protection, equity [5], and feasibility of implementation [6], all of which need to be considered in the context of health financing mechanisms [7].

For transparency and clarity in the interpretation of complex considerations, the use of quantitative methods is recommended [8] to formally weigh evidence in support of different interventions. Examples of such methods include league table [1] and constrained optimisation [9,6] approaches usually established to maximise population health from limited resources, and multi-criteria decision-analysis (MCDA) [8,10,11] which seek to rank interventions on the basis of multiple criteria, including those on which quantitative evidence is not readily available or hard to obtain. They additionally include analyses which focus on priority setting for resource allocation in health [12,13]. These quantitative analyses are then feed into the wider deliberative process of adopting an HBP.

These quantitative inputs into HBP design, however, are faced with a number of methodological limitations. The incidence of relevant diseases and medical conditions is assumed to be independent, such that important comorbidity effects and interactions in the availability of interventions are not captured, as well as possible shifts in the epidemiological context over time. Issues of double-counting also affect the estimated return from the inclusion of different interventions based on CE evidence alone [14].

In addition, these analyses use CE estimates which necessarily include intrinsic assumptions about the probability of access and uptake and probability of service implementation, which happened to hold true for the country and period in which the data were collected (although attempts have been made, for example, through WHO-CHOICE, to estimate the cost-effectiveness of interventions at different levels of coverage [15]). Evidence for such analyses must indeed be drawn from disparate sources (see the "Disease Control Priorities Network" [16] and the "Tufts Cost-Effectiveness Registry" [17]) which may rely on evidence obtained from diverse geographical and temporal contexts, as well as different methodologies. Even in the case of current, country-specific CE estimates, the intrinsic nature of such context-dependent assumptions limits the ability of CE evidence-based analyses to consider predicted or hypothetical changes—for example, if seeking to evaluate an HBP strategy in conjunction with programmes aiming to remove financial barriers in access and uptake—hence severely restricting the range of scenarios these analyses can evaluate. CE estimates may also fail to capture prediagnosis costs [18].

Finally, while methods are available to account for the estimated uncertainty in the available CE evidence [19,20], a complete lack of relevant evidence on a significant number of interventions means that these must be excluded *a priori* from quantitative analysis [16], although this issue can be overcome through participatory and deliberative processes (including the use of MCDA [18]).

One of the countries that relies on CE evidence analyses as input to its HBP deliberative process is Malawi (see page 49, [21]). The Government of Malawi (GoM) is indeed committed to meeting Sustainable Development Goal 3.8 of achieving Universal Health Coverage (UHC) [21]. With a total combined annual healthcare expenditure from government and donors of $ 39.8 per capita per year [22], the financial resources available to deliver this commitment are extremely scarce. The GoM has therefore taken a proactive approach to resource prioritisation, and its Health Sector Strategic Plan (HSSP) I (2011–2016) [23], II (2017–2022) [24], and III (2023–2030) [21] were all based upon HBPs informed by such LCOA and MCDA (as referenced in each HSSP).

In this work we introduce, for the example of a Malawi-specific setting, a novel approach to the quantitative evaluation of potential HBPs based on the individual-based epidemiological model *Thanzi La Onse* (TLO). The TLO model is designed to comprehensively forecast, specifically in the context of Malawi, the health outcome achievable by any HBP while:

- self-consistently modelling the incidence of all relevant medical conditions and comorbidity effects, as well as risk-factors and demographic evolution of the population, over the period in which the HBP is to be adopted;

- making explicit—and hence easily modifiable—assumptions about the probability of service delivery and of access and uptake among those who would benefit from the intervention, as well as their health-seeking and treatment- adherence behaviours;

- enforcing healthcare-resource constraints.

Crucially, this country-specific predictive tool therefore addresses the limitations in CE evidence-based analyses outlined above, and as such can provide a useful and additional input to the country's HBP generative process.

In particular, our analysis evaluates a limited set of possible HBPs to establish the relative reduction in the overall health burden that could be achieved through different prioritisation strategies, and to showcase how model-based analyses can offer an important insight into the factors that make individual HBPs particularly successful or unsuccessful in achieving this outcome. The HBP performance evolution over time is also evaluated.

## 2. Method

### 2.1. Overview of analytic approach

We use the TLO simulation (1 www.tlomodel.org) [25] to simulate the health impact that may be realised in Malawi under alternative, credible formulations of the HBP when ensuring that (a) services delivered by the healthcare system are constrained by a realistic estimate of the time available from its workforce, (b) services take a realistic amount of time to be delivered, and (c) all other aspects of the healthcare system are analogous to the status quo (including clinical practice and availability of consumables, beds, and equipment).

Each possible formulation of the HBP is modelled as a statement of the relative priority that is placed on a particular service, or on a particular group of patients (e.g. currently pregnant, or living with human immunodeficiency virus (HIV)). High-priority services or patients will be attended first, such that lower-priority ones will only be attended if healthcare workers still have time available to do so, resulting in a healthcare system that *preferentially* delivers certain services over others. Although exclusion of services will in some cases be considered, this statement of priority is a more nuanced version of the "in or out" decision for each service that has been used in other analyses (see the discussion on this point in section 4); for consistency, we will refer to the HBPs considered in this analysis as "prioritisation policies" from now on.

The analysis is agnostic to how such gradations of "priority" are to be realised in practise (e.g. reserved times in clinics for certain patients, dedicated facilities for particular services, the addition or removal of certain services entirely from public sector healthcare). Instead, it is designed to reveal the type of outcome that such operational decisions may aim to realise.

The TLO simulation starts on 1 January 2010, and is calibrated to data in the period 2010–2020. Input from the Malawian Ministry of Health, as well as from several clinicians working in the country, was sought and accounted for at all development stages of the model, alongside a large number of data sources (see [25]). The analysis conducted in this work assumes the prioritisation policies evaluated are implemented from 1 January 2023 onward and extends to the following 20 years until 31 December 2042, inclusive, to capture the long term effects of the policies considered. As the output produced by the simulation is broken down by year, this allows us to evaluate the performance of the policies on any time period up to 20 years, as discussed in section 3.3. The initial population size adopted (in 2010) was 100,000 individuals, and we performed 10 simulations per scenario considered, each with independent random draws; both numbers were found to be large enough to give stable estimates of the mean and variance of the health burden obtained under each cause included in the simulation over independent realisations. The results reported are scaled to the true population size in Malawi in 2010 of 14.5 million individuals.

### 2.2. Resource constraints

The incidence of disease and disability among individuals in the TLO simulation, combined with assumed models for health-seeking behaviour (see model's overview paper [25] and references therein), determine the number of people seeking specific healthcare system interactions (HSIs) at each of the healthcare facilities modelled in the simulation on any given day. Such HSIs can be both preventative and curative in nature, while the impact of a variety of non-medical initiatives (such as outreach programmes on condom use for HIV prevention and contraception, use of insecticide treated bed nets, etc) is captured through its effect on the probability of individuals contracting relevant diseases and subsequently seeking care. The simulation accounts for the possibility of reinfection among individuals who have recovered from the disease before, and considers the impact of prior treatment and/or natural immunity on both the likelihood of reinfection and potential outcomes.

Each requested HSI, in return, is associated with an assumed time requirement from one or more types of healthcare worker (HCW) [26], as informed by the Human Resources for Health Strategic Plan (HRH SP) 2018–2022 [27]. (For example, we assume that a standard outpatient appointment for a patient over five years old at facility level 1a would require 27 minutes of clinical officer time, 18 minutes of nurse time, and 9.5 minutes of pharmacist time). This results in a realistic and comprehensive demand for each medical cadre–including for community health workers–taking place on a facility-by-facility basis every day as a result of all medical conditions included in the simulation.

On this demand for HCWs time we base the resource constraint enforced in this analysis during the period of interest (from 1 January 2023 to 31 December 2042): during this period, daily capabilities assumed at each simulated facility are capped, reflecting up-to-date data on human resources for health currently available in the country (see [26] for a detailed overview of human resources for health modelling in the TLO simulation). Furthermore, the healthcare system is only allowed to dispense treatments until such capabilities are exhausted in a *rigid* mode of service delivery, characterised by the following assumptions:

- HSIs can only be delivered in the assumed expected duration given the treatment type, and cannot be shortened by medical practitioners to accommodate a high demand for services on the day.

- HCWs cannot perform overtime that would extend their total assumed capabilities, except to complete the last HSI of the day.

- No task shifting between medical cadres is allowed.

- If any of the HCWs required for a given HSI have exhausted their capabilities for the day at the facility where the patient initially sought care, the patient will not be able to receive the requested service on that day.

- If any of the consumables required are not available due to stock-outs [28], the patient may receive an alternative treatment during that HSI, request a repeat visit at a later date, or default from care ([29].

Given the rigid mode of service delivery and a limit on daily capabilities, services requiring the same HCW at the same facility level will therefore be represented as being in direct competition with each other. The *order* in which competing HSIs are delivered will therefore play an important role in determining which services will be successfully delivered before capabilities exhaustion, and which will have to be postponed, as discussed in the next section.

## 2.3. Prioritisation of services and patients

In our analysis, the competition for constrained HCWs' time outlined in the previous section is mediated via a prioritisation approach: each patient seeking treatment is assigned a priority level, which determines the order in which they will be seen at the healthcare facility they have attended. For patients with an equal level of priority, the order in which they will be seen is determined by the date on which they first started to seek care—such that patients who have been seeking care for longer should receive treatment first—or, if that date is also equal, randomised.

At the start of the day, HSIs are progressively delivered under the rigid healthcare system assumption (see section 2.2) in the specified order, as illustrated in Fig 1 (left plot). Patients who did not receive treatment on the day they initially sought care may seek care again at the same facility the following day for a maximum of seven days (see section 4 for a discussion on

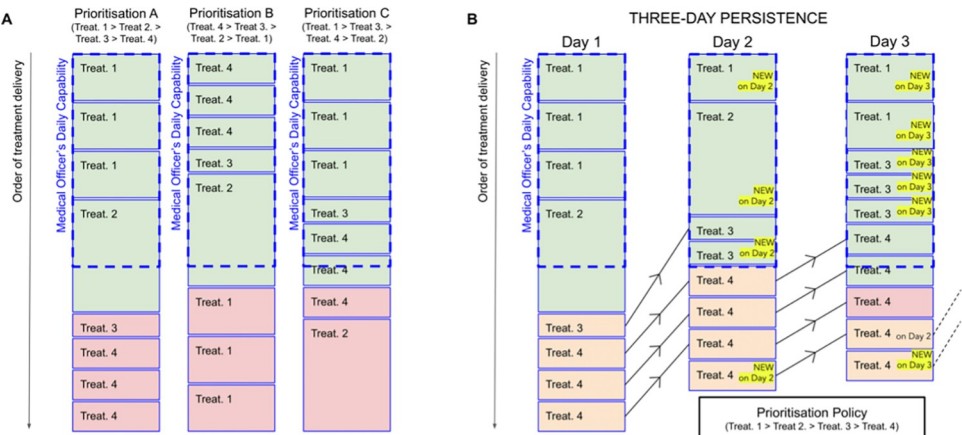

**Fig 1.** *Left plot (A)*: Illustration of treatment delivery under a "rigid healthcare system" assumption (see section 2.2) when implementing three different prioritisation policies A, B, C. In this example, the healthcare system can provide four types of treatments (Treatments 1, 2, 3, and 4), which require different amounts of time from a medical officer, as illustrated by the height of each treatment box. On this day, eight treatments are requested in total (three Treatments 1 and 4, and one Treatment 2 and 3). The prioritisation-policy at the top of each column specifies the order in which treatments would be delivered under that policy (where">" signifies that the treatment on the left is prioritised above the one on the right), while the dashed blue box shows the total daily capability of the required medical officer available. As a result of the resource constraint, only certain treatments can be delivered (shown in green) on that day before capabilities are exhausted (recall that overtime is allowed to complete the last treatment of the day), while all remaining ones (shown in red) will have to be postponed to a later date. Therefore, the diagram illustrates how, as a result of the adoption of different prioritisation policies, different types of treatments are preferentially delivered under a resource-constrained, "rigid" healthcare system. *Right plot (B)*: Diagram illustrating how health-seeking persistence is implemented under a "rigid healthcare system" assumption and constrained daily capabilities. Individuals who do not receive treatment on the first day (highlighted in orange) can seek care again on following days until they have exhausted the assumed maximum number of attempts (three in this example for illustrative purposes, but seven in the simulations). If, after reaching the maximum number of attempts, they still have not succeeded in receiving treatment, they will default from care entirely, and that treatment will never be delivered (as in the case for one of the Treatments 4, highlighted in red). Note that the order in which treatments are organised on subsequent days is determined both by the prioritisation-policy adopted and, for treatments with the same priority, by the date in which care was first sought, as discussed in section 2.3.

this assumption). When patients return to seek care for the same condition, we assume they should expect to receive treatment before patients with the same level of priority who started to seek care more recently, but after new patients with a higher level of priority, as illustrated in Fig 1 (right plot).

Therefore, a higher level of priority increases the probability of an individual being able to access care before medical capabilities are exhausted for the day. The result will be a healthcare system preferentially delivering certain services competing for the same resources over others, but which can still perform lower priority HSIs should capabilities still be available. In practice, a number of other factors will influence the relative rate of service delivery, as discussed in detail in S1 Appendix.

Criteria adopted in the allocation of priority to patients seeking treatment either invoke the type of treatment they are seeking, properties intrinsic to the individuals themselves (which may qualify them to be classified as members of a vulnerable category), or both. For example, the highest priority could be assigned to patients seeking treatment for pneumonia specifically; to all children under five years of age seeking any type of treatment; or to all children under five years of age seeking treatment for pneumonia specifically. This approach is motivated by previous literature and practice advocating for a transparent method of prioritisation in the context of insufficient resources to meet the needs of the entire population in need.

The possibility of considering patient characteristics as a discriminating factor in resource allocation is a unique advantage of prioritisation-policy evaluations over traditional conceptualisations of the HBP, where the treatment type has been the only criterion for prioritisation, though sometimes specified by patient type. The vulnerable categories currently considered in the simulation are children (five years of age or under); pregnant women; patients diagnosed with tuberculosis (TB); and patients diagnosed with HIV, which define characteristics constituting important risk factors for a multitude of medical conditions. While many more categories of clinically vulnerable patients could be considered, this represents an initial attempt to establish whether the prioritisation of patient characteristics rather than service type could become a useful simplified strategy in the effort for more efficient resource allocation.

The HSIs modelled in the TLO simulation are grouped into 82 types of treatment that patients could receive for their conditions [25]. These treatment categories are roughly equivalent to those adopted by the Ministry of Health in its HSSP Health Benefits Package [21] (see S2 Appendix for a detailed description of how the TLO treatments map to those of the HBP currently adopted in Malawi), and it is therefore at this level—rather than at that of individual HSIs—that we consider a prioritisation strategy.

We define a prioritisation-policy as evaluated in this analysis, as the allocation of a priority level to *each* of the 82 treatments modelled by the TLO simulation, *and* the specification of whether each treatment is further eligible for a fast track to a higher priority level if the individual seeking it belongs to one or more of the specified vulnerable categories. Policies may also choose to exclude entirely the provision of certain treatments. By default—and therefore beyond the scope of individual policies—we assume that the highest possible priorities are reserved for treatments classified as emergencies, with emergencies relating to children being further prioritised over adults (see overview paper [25] for a discussion on which treatments are classified as emergencies in the TLO model). Individual prioritisation policies can therefore only determine how non-emergency treatments should be prioritised. The prioritisation policies that will be evaluated are discussed further in the next section.

## 2.4. Prioritisation policies considered

In our analysis, we focus on evaluating a limited set of prioritisation policies. These policies, summarised in Table 1, reflect either:

- A complete lack of prioritisation strategy (the *No Policy* (NP) policy);

- The outputs of traditional HBP-design methodologies outlined in section 1, namely the MCDA-informed HBP adopted under HSSP-III [21] (the *HBP from Health Sector Strategic Plan III* (HSSP-III HBP) policy) and the output of a Malawi specific LCOA [18] (the *Naive LCOA* (LCOA) policy);

- Widely held presumptions—among policy-makers, funding agencies, and relevant literature—around what kind of services should be prioritised in order to obtain a reduction in health burden in Malawi (the *Vertical Programmes* (VP), *Reproductive, Maternal, Neonatal, and Child Health* (RMNCH), and *Cardiometabolic Disorders* (CMD) policies);

- A new approach to service prioritisation centred around patient characteristics only rather than treatment type (the *Clinically Vulnerable* (CV) policy);

The individual motivation of each prioritisation-policy and a detailed breakdown of the priority levels assigned to each treatment and vulnerable category under that policy is discussed in more detail in S3 Appendix.

**Table 1. Summary of prioritisation policies considered for evaluation.** A more extensive description and motivation of each policy is included in S3 Appendix.

| Name of policy | Acronym | Description | Motivation |
|---|---|---|---|
| **No policy** | NP | Equal priority for all non-emergency HSIs. | Benchmark in evaluation of other prioritisation policies. |
| **Naive LCOA** | LCOA | Mapping to interventions included and excluded by the prioritised list of interventions emerging from the linear constrained optimisation analysis (LCOA) carried out to provide initial technical input into the design of HSSP-III HBP. The LCOA was set up to maximise the net health benefit from interventions given Malawi's health system resource constraints and demand constraints, assuming a cost-effectiveness threshold of $65.8/DALY averted. We consider the LCOA output which does not consider donor constraints and therefore "naively" maximises net health benefit given the cost-effectiveness evidence on interventions available in 2021 and assuming perfect fungibility of the health sector budget across disease programs. | Analyse performance of the LCOA output. |
| **HBP from Health Sector Strategic Plan III** | HSSP-III HBP | Mapping to interventions included and excluded by the HBP adopted by the GoM under HSSP-III. The interventions were prioritised based on initial input from the LCOA, augmented by MCDA incorporating the following factors—severity of illness, effectiveness, poverty reduction, vulnerable populations, and level of care. | Analyse performance of the current HSSP-III HBP. |
| **Vertical Programmes** | VP | High priority to all malaria, HIV, TB, and immunisation services. | Assess impact of only prioritising donor-sponsored programmes. |
| **Reproductive, Maternal, Neonatal, and Child Health** | RMNCH | Perinatal appointments receive the highest-possible prioritisation, followed by appointments relevant to children's health (acute lower respiratory infections, diarrhoea, EPI appointments, measles, schistosomiasis, and under-nutrition), and by contraception appointments. All other treatments have low priority. | Assess impact of prioritising RMNCH areas of health above all others. |
| **Cardiometabolic Disorders** | CMD | High priority to all treatments related to cardiometabolic disorders. | Assess opportunity cost of early intervention in this area of health. |
| **Clinically Vulnerable** | CV | Patients who classify for at least one of the "clinically vulnerable" categories considered (five years of age or under; pregnant women; HIV-diagnosed patients; or TB-diagnosed patients) are always assigned a high level of priority, regardless of the type of service they are seeking. All other patients are assigned a lower priority. | Assess impact of prioritising patient characteristics rather than treatment types. |

In evaluating this limited set of possible prioritisation policies, we seek to establish, first of all, whether any of them can indeed lead to a reduction in the overall health burden incurred through mediation of resource access alone, without requiring any expansion of available healthcare capabilities (section 3.1). Secondly, we seek to gain insight into the factors that make individual policies particularly successful or unsuccessful in achieving this outcome (section 3.2). Finally, we evaluate the evolution of policy performance over time (section 3.3).

## 3. Results

The overall health burden incurred under each prioritisation-policy between 2023 and 2042 (the period in which policies are adopted and a *rigid* healthcare system assumption is enforced) is quantified by disability-adjusted life years (DALYs) assuming an average life-expectancy [30] of 70 years, and adopting disability adjustment factors from [31].

### 3.1. Overall health outcomes

In Fig 2 we show the total DALYs incurred overall between 2023 and 2042 (inclusive) under different prioritisation policies. Three of the proposed policies—namely LCOA, CV, and VP—appear to successfully provide an overall benefit compared to the NP scenario, with LCOA providing the largest gain. This seems to suggest that enforcing a prioritisation strategy in a resource-constrained environment can indeed result in an overall gain in health (as high as a ∼ 8% reduction in total incurred DALYs) over an equal access approach (NP), crucially without requiring any expansion in existing capabilities. On the other hand, the implementation of an RMNCH policy led to a significant increase in DALYs incurred, while HSSP-III HBP and

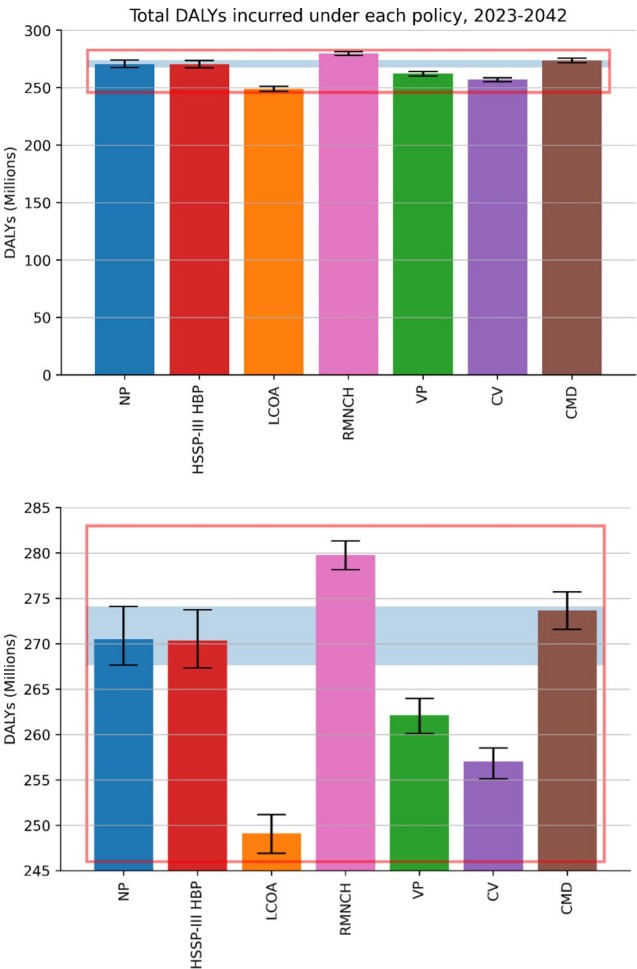

**Fig 2.** *Top plot*: Total DALYs incurred overall (between 2023 and 2042 inclusive) under each policy considered. Error bars show the 95% confidence interval (CI), with the shaded blue region extending the NP ones to facilitate comparison. *Bottom plot*: Zoom-in of the range highlighted in red in the top plot.

CMD did not result in significant benefit, suggesting that careful considerations should be made on how services are prioritised.

The performance of these policies over the entire 20-year period will be shaped by a combination of their performance in different key areas of health and the relative contribution of these to the overall health burden, which may vary during the simulated period.

In S4 Appendix, we discuss in detail the relative performance of the evaluated policies in some of the leading causes of DALYs (acquired immune deficiency syndrome (AIDS), lower respiratory infections, neonatal disorders, malaria, TB, and measles), by comparing both the health burden incurred in those areas of health and the rate of delivery of relevant services. In Fig 3, on the other hand, we summarise the mean DALYs incurred due to all simulated causes, where these have been ranked from the highest to lowest contributors over the entire 20-year period in the case of a NP policy, in order to better understand how the relative performance of the policies in different areas of health shapes the relative performance of the policies overall. Notice that DALYs incurred under the label of "Other"—which are included in the simulation to account for causes of DALYs not explicitly modelled, and whose incidence is calibrated

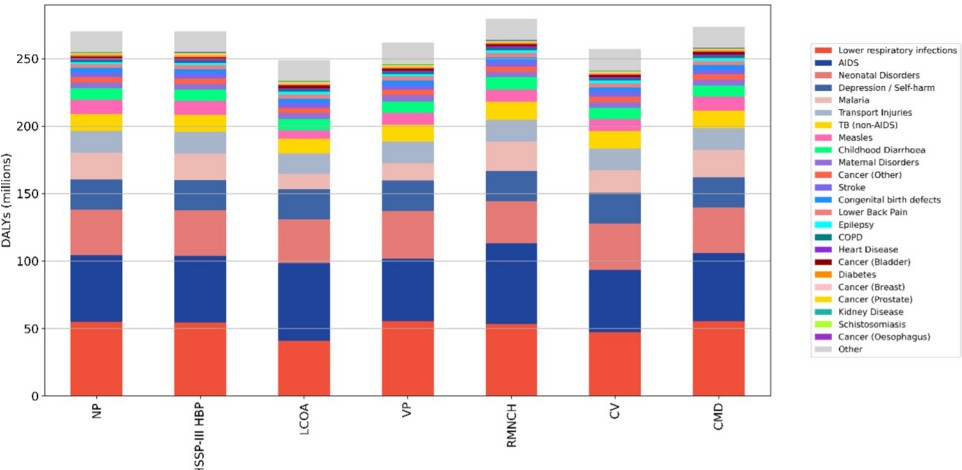

**Fig 3. Mean DALYs lost overall between 2023 and 2042 (inclusive) under different policies broken down by contributing causes, where the causes (excluding "Other" causes, which are added at the end) have been ranked from the highest to the lowest contributing under the NP policy.**

to data and not subject to resource constraints—are not included in the ranking and simply added at the end.

From this figure we see that LCOA makes the most significant gains in lower respiratory infections, the leading cause of DALYs over the entire 20-year period, while performing quite poorly in HIV/AIDS, the second. VP, on the contrary, performs quite well in HIV/AIDS, but incurs significant losses due to lower respiratory infections, which are deprioritised under this policy. By prioritising patient characteristics rather than specific services, CV achieves a more balanced distribution of resources in these two key areas of health, and hence a slightly better performance from these two combined causes. RMNCH is the worst performing policy at this stage: while its poor performance under HIV/AIDS is not unexpected, the prioritisation of perinatal care above pneumonia treatment for children means that most resources are allocated toward the former before they can reach the latter. The higher rate of perinatal services delivered, however, does not result in a significant reduction in DALYs incurred due to neonatal or maternal disorders (see S4 Appendix for a detailed discussion), making the RMNCH policy the worst-performing policy overall already by this stage.

Once lower respiratory infections, HIV/AIDS, and neonatal disorders are accounted for, LCOA and CV policies achieve a similar performance, with the latter showing, on average, only a marginal advantage. While DALYs incurred due to depression (fourth cause) and transport injuries (sixth cause) are virtually unaffected by the choice of policy, LCOA is, however, able to outperform CV in other important areas of health, namely malaria, TB, and measles, due to its ability to provide a higher volume of services in those areas (see S4 Appendix).

HSSP-III HBP performs remarkably similarly to the NP in most high-profile areas. Although, similarly to the LCOA, it deliberately excludes certain treatments from its provision (primarily concerning cancer care), it does not exclude many. Furthermore, unlike the LCOA, it does not include a prioritisation strategy for the ones that are included. It is therefore not surprising that the performance of the two policies is quite similar, given that both the contribution of cancer to overall DALYs and request for treatment is quite low, and hence does not significantly impact the competition for services. Similarly, the currently low contribution of cardiovascular diseases to the total DALYs incurred means that, for the CMDs policy, the preferential delivery of CMD-related treatments does not result in significant improvements in

overall outcomes, and only leads to marginally worse outcomes in other areas of health as services targeting more prominent causes of DALYs are deprioritised.

This clearly illustrates how the overall performance of different policies is shaped by the combination of their gains and losses in different areas of health. But what enables certain policies to achieve better outcomes than others overall?

## 3.2. What makes a prioritisation strategy successful?

Comparing the relative performance of the set of prioritisation policies evaluated allows us to identify the following factors as playing an important role in determining whether they are more or less likely to reduce the overall health burden.

**Striking a balance between the inclusion of essential services and an effective resource-access mediation strategy.** A key factor determining the relative performance of different policies is whether vital services in key areas of health could be delivered to a satisfactory degree. Two main aspects of policy design were responsible for determining to what extent this could be achieved: first, whether policies sufficiently prioritised or included those services in their healthcare provision in the first place, and second whether they ensured that enough resources could be safeguarded to provide those services.

Among clear examples of the former is the de-prioritisation of all pneumonia services under the VP policy, which led to significant losses due to lower respiratory infections under that policy. LCOA, similarly, incurred a significant excess of HIV/AIDS-caused DALYs due to its complete exclusion of HIV treatment from its provision (which was motivated by the low CE of antiretroviral treatment found by the most recent, though outdated study available at the time the LCOA was conducted [32]), which could not be contained by the prioritisation of HIV prevention and testing services. The CV policy, however, by prioritising vulnerable categories of patients rather than focusing on specific services, was instead able to ensure that all key services—identifiable as such because requested by vulnerable patients likely to incur significant DALYs if left untreated—were assigned a high priority in any area of health.

Ensuring key services are correctly identified as such and assigned a high priority by a policy, however, is not sufficient to ensure that these can be delivered to a satisfactory degree. Instead, insufficient mediation of resource-access among high-priority appointments meant that, in some cases, even policies that placed high priority on key services at times struggled to deliver them. For example, while most patients requiring inpatient pneumonia appointments would have been highly prioritised under a CV policy (as pneumonia primarily affects and leads to healthcare seeking in children), the LCOA was able, in most cases, to deliver a much higher number of such appointments, even if these were actually assigned a low priority under this policy. (A further discussion of the factors driving the competition for constrained resources under different policies can be found in S1 Appendix). As a result, LCOA was able to make more significant gains in this area of health than the CV policy (see S4 Appendix).

The reason the LCOA was able to deliver a higher rate of such services is that it is both more selective in the number of services that qualify for high-priority—for example through the de-prioritisation of HIV treatment, which instead classifies as a high-priority service under most policies—and further mediates resource competition for low-priority services by deliberately excluding the delivery of certain treatments altogether. The more targeted selection of high-priority appointments and a more nuanced prioritisation strategy (which effectively takes place at three levels, namely high priority, low priority, and excluded treatments) meant that more resources were available to deliver TB, malaria, and measles treatments. On the contrary, the CV policy, although very efficient at ensuring no key services are fully deprioritised if

requested by a vulnerable patient, is likely not selective enough in establishing which services truly warrant a high priority in those cases.

**Accounting for quality of care and consumables stock-outs.**   The assumption that a higher rate of key-service delivery directly translates into a reduction of the health burden can be significantly undermined if the quality of the care delivered and the availability of necessary consumables are compromised. This could not be illustrated more clearly than in the case of the RMNCH policy: although this policy allocates a huge amount of resources towards perinatal care, its relative gains in neonatal and maternal health compared to other policies are marginal, due to a combination of low availability of key consumables and skilled medical professionals (see [33] and S4 Appendix for a detailed discussion). The opportunity cost of allocating resources towards these services is therefore extremely large, as illustrated by the fact that the RMNCH policy struggles to deliver services in any other areas of health, and as a result is the worst performing policy overall.

**Addressing risk factors.**   The ability of policies to effectively address risk factors plays a crucial role in areas of health where they are an important driver of DALYs outcome. The clearest example of this is offered by measles (see S4 Appendix): lack of vaccination constitutes an important risk factor for the onset of related complications. As shown in Fig F in S4 Appendix, the strongest determinant of the health burden due to this disease is not the rate of measles treatment delivery under different policies, but rather that of measles vaccination. The CV policy is one that, more than any other, is capable of effectively using preventive strategies, as it ensures that the most vulnerable individuals have prioritised access to all services that could reduce risk factors and therefore future health outcomes, contributing to the success of this policy overall.

## Dynamic effects on the incidence of diseases over the implementation period

One of the difficulties in predicting the impact that the inclusion of a given intervention will have over the implementation period is related to the uncertainty about how the incidence of different diseases and medical conditions will evolve during that time. This is due both to uncertainty around the demographic growth paths the population could take—which, as shown in S5 Appendix, is linked among other things to the inclusion or exclusion of contraceptive and family-planning services—, the incidence of comorbidities and other risk factors, and nonlinear effects in the spread of infectious diseases (as clearly seen in the case of HIV, discussed in detail in S4 Appendix). The TLO simulation is able to capture these effects self-consistently, which highlights how the deprioritisation of infectious disease treatment (as in the case of HIV treatment for LCOA) can lead to a rapid escalation of incurred DALYs. This leads to important temporal effects in the evaluation of different prioritisation policies, which will be discussed next.

## 3.3. The relative performance of different prioritisation policies at different stages of implementation

The dynamic evolution of different diseases and medical conditions over time leads to their relative contribution to the overall health burden changing significantly over the period of implementation of the policies. This is illustrated in Fig 4, where we show how the total health burden is shaped by the evolution of the ten leading causes of DALYs—which alone account for $\sim 85\%$ of all DALYs incurred—for the benchmark case of the NP policy: while HIV/AIDS dominates by far the health burden incurred initially, this is overtaken by lower respiratory

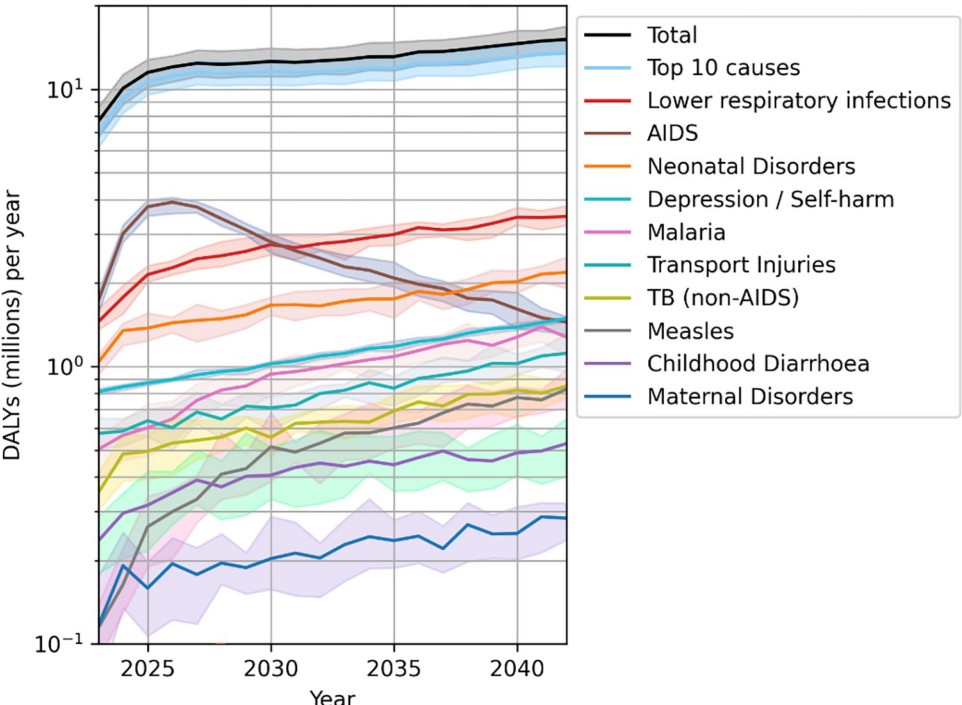

**Fig 4. Evolution of the ten leading causes of DALYs over the period considered for the benchmark case of the NP policy, ranked by their overall contribution in the 2023–2042 period.** The black line shows the total DALYs incurred from all causes, while the light blue line shows the DALYs contributed by the top ten causes.

infections as the leading cause of DALYs by 2031, with neonatal disorders closely following the same rising, demographically-led trend.

In the previous sections we have seen that the policies evaluated have relative advantages in different key areas of health, due to their individual strategy in preferential resource allocation towards different services, and their more or less effective mediation of resource competition. As the relevant contribution of these areas of health to the overall health burden varies significantly over time, we should therefore expect the relative overall performance of these policies to vary greatly at different stages of implementation.

This is clearly illustrated in Fig 5, where we show the year-by-year evolution of total DALYs incurred in the TLO simulation under the different policies. We see that the assumed transition to a rigid healthcare system in 2023 results in a rapid increase in incurred DALYs across all policies, as access to services becomes resource-constrained, but that the rapid rise in DALYs is particularly significant for two policies, namely RMNCH and LCOA. This is consistent with HIV/AIDS being by far the leading cause of DALYs in the initial period (see Fig 4), and the RMNCH and LCOA being the worst performers in this area of health (see S4 Appendix). While LCOA later recovers due to its gains in other key areas of health such as lower respiratory infections, malaria, TB, and measles, the RMNCH policy performs poorly by dedicating most of its effort towards perinatal care with little benefit, incurring huge losses in other areas of health.

CV, on the other hand—whose overall performance is not as good as that of LCOA, although still better than the NP—avoids the excess in health burden initially incurred by LCOA, providing a more marginal but consistent health gain on a year-by-year basis, through a more balanced approach in services allocation by prioritising those most at risk, rather than specific services. VP, which contains the initial spike in HIV, also manages to maintain a mean

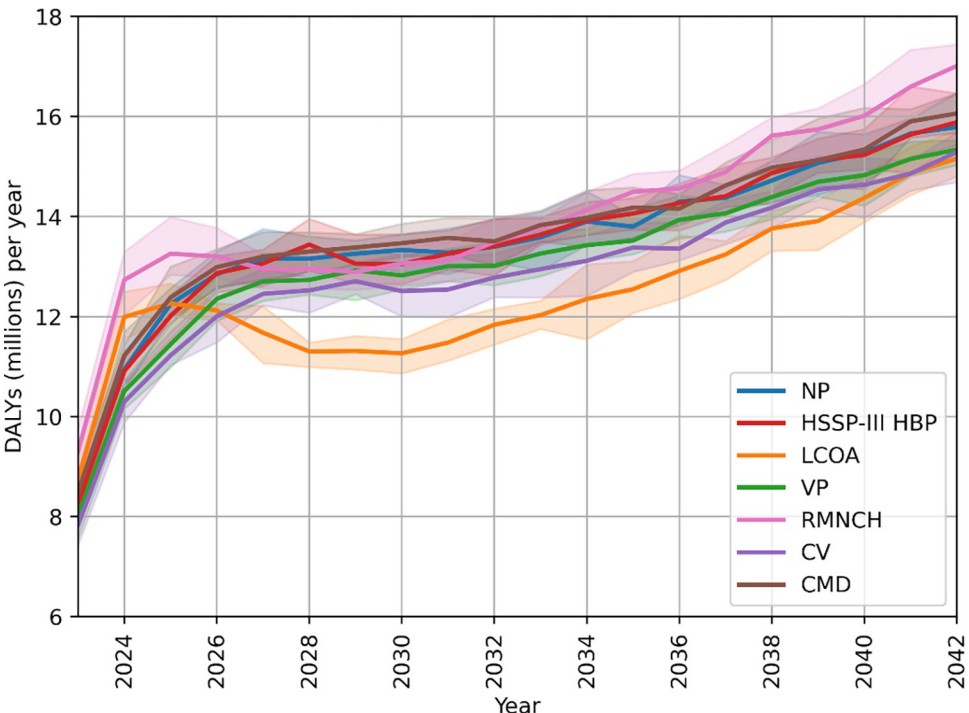

**Fig 5. Yearly breakdown of DALYs incurred yearly under each policy considered.** Error bars show the 95% confidence interval (CI).

performance consistently below that of NP, although it incurs on average an additional $\sim 0.2$ million DALYs a year compared to CV. HSSP-III HBP and the CMD policy closely follow the NP trend bringing no overall benefit—although no apparent disadvantage either—at any point in the simulated period.

Therefore, while LCOA is the best performing policy overall when taking a simple cumulative contribution of DALYs over the entire 20-year period, it performs worse than a NP approach in the first few years. If the event horizon over which policies were to be evaluated was chosen to be much shorter the relative gains achievable under each policy would look much different. In Fig 6 we show an example of DALYs incurred over event horizons of three and five years from the time when policies are initially adopted. We see that the adoption of the LCOA policy—the policy that performs best in the overall 20 year period—would either lead to a worse health outcome in the shortest term considered, or to one at best comparable with that of a NP approach over the first five years. CV and VP, on the other hand, appear to perform consistently better than the NP policy irrespective of the event horizon considered, with CV offering consistently better gains.

Differences in the temporal performance of the prioritisation policies can have important consequences for which prioritisation strategy policymakers may choose to adopt. In practice, discounting may alternatively be considered, however there are debates as to how this can most reliably be undertaken [34,35]. For this reason, we have shown undiscounted outcomes in the analyses presented.

## 4. Discussion

In this analysis, we used the *Thanzi La Onse* (TLO) simulation to evaluate the health burden that should be expected in Malawi over the period 2023–2042 assuming the healthcare system

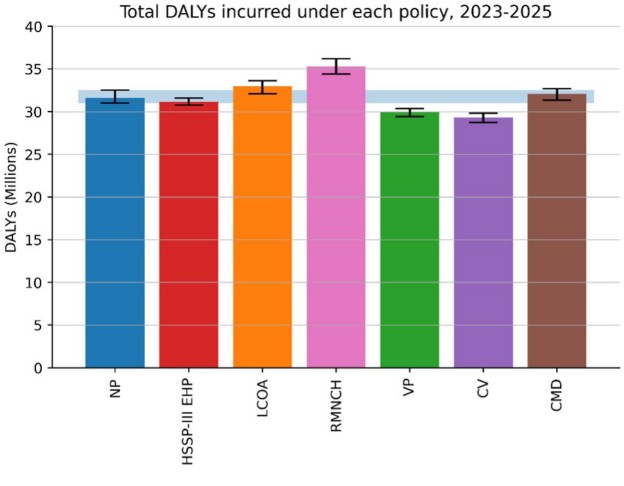

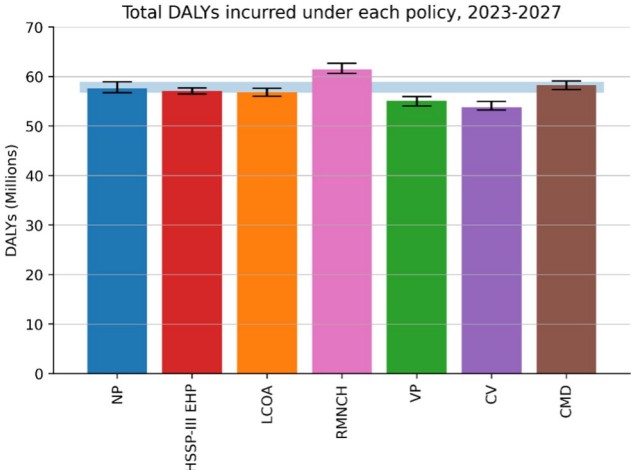

**Fig 6.** Overall DALYs incurred under different policies during the initial three years (top figure) and five years (bottom figure) after the adoption of the policies. This illustrates that an implementation period of five years or longer is necessary for the LCOA to provide a significant benefit over no prioritisation strategy at all.

is strictly human-resource-constrained and adopting a number of service prioritisation strategies, which are summarised in Table 1. The TLO simulation is able to self-consistently capture changes in the rate of medical-service delivery resulting from enforcing a prioritisation strategy in a resource-constrained healthcare system. It is also able to capture subsequent changes in cause-specific DALYs lost, which not only reflect the different rates of service delivery, but further capture incidence of comorbidities and risk factors, health-seeking behaviour, referral pathways, realistic quality of care, and consumables availability, all calibrated to Malawi-specific data. The key findings of our analysis are:

- The adoption of different prioritisation strategies (or "prioritisation policies") in the delivery of services leads to significant differences in the overall health burden incurred over the simulated period. In particular, we found that several of the evaluated prioritisation policies (LCOA, CV, and VP) achieve a significant reduction in overall DALYs lost over the 20-year period (2023–2042) over a NP in a resource-constrained healthcare system, without requiring an expansion of existing capabilities.

- The most successful policy in reducing the incurred health burden over the entire 20-year period was LCOA, under which the more selective choice of high priority appointments and nuanced prioritisation strategy (which distinguished between high priority, low priority, and excluded services) resulted in resources being more effectively allocated to higher-impact services.

- The year-by-year performance of different policies however varied greatly over the simulated period, with the LCOA policy actually incurring an excess in DALYs over the initial three year period compared to the NP policy, largely resulting from its exclusion of HIV treatment from its provision. This suggests that the choice of different time horizons or the inclusion of discounting would significantly impact the evaluation of the selected policy.

- The policy that led to 5% net health benefit compared to the NP policy while simultaneously ensuring that the health burden was reduced consistently over the NP on a year-to-year basis was the CV policy, which achieved marginal improvements in all key areas of health by prioritising vulnerable patients rather than specific services.

This TLO-based evaluation was able to address several of the limitations affecting current methods for HBP design listed in section 1, and demonstrated how a dynamic modelling of both the incidence of disease and competition for limited resources plays a crucial role in the evaluation of different prioritisation policies. A number of limitations, however, do affect our analysis, as discussed below.

### 4.1. Rigid healthcare system assumption

The TLO simulation, for the first time, allowed us to place a realistic constraint on the amount of health that the Malawian healthcare system is able to deliver based on data-driven estimates of human resources for health as well as time requirements for different types of appointments. Modelling assumptions required to enforce this in practice were summarised under the rigid healthcare system approach (see section 2.2); some uncertainty around these assumptions, however, exists in reality. [26] suggest medical practitioners in Malawi, on average, are likely to work overtime and/or adjust the duration of appointments to accommodate a high volume of patients per day. In this case, our analysis may therefore overestimate the health burden incurred when enforcing resource constraints in the analysis.

This is perhaps most visible in the case of HIV/AIDS-caused DALYs, which incur a sharp increase in 2023 following the transition to a rigid healthcare system assumption (see Fig B in S4 Appendix) as the healthcare system struggles to meet the demand for HIV treatments. Only ∼35% of all HIV treatment requests between 2023 and 2024 are in fact delivered under the VP policy, which has the highest rate of HIV treatment delivery. MPHIA data [36], on the other hand, shows that most people (97.9%) diagnosed with HIV in Malawi are receiving anti-retroviral therapy. This discrepancy suggests that, in reality, healthcare professionals are able to provide such services through a combination of overtime and/or reduction of appointment duration. Our results therefore confirm the extent to which the healthcare system relies on such practices to provide a much higher volume of service delivery than should be expected, and hence the benefits that could result from an increased efficiency in the delivery of HIV services in reducing the workload of HCWs. (An important exception to such considerations should be noted in the case of the LCOA which, unlike any other policy, excludes HIV treatment from its provision. For this policy, the HIV/AIDS-caused DALYs spike is completely independent of our rigid healthcare system assumptions, and instead robustly highlights the extend to which the de-prioritisation of HIV services in the country would result in a rapid increase in HIV/AIDS-caused morbidity and mortality. Since the performance of the LCOA

policy for the case HIV/AIDS would be unaffected by allowing some overtime and/or shortening of appointment duration to take place, while that of other policies would be improved, this suggests that the performance of the LCOA relative to that of other policies in the initial HIV/AIDS-dominated period—discussed in section 3.3—may be even less advantageous than currently estimated).

It is, however, plausible to assume that the practice of rushing of appointments, overtime, and task shifting may in the long term affect the quality and quantity of the care delivered, for instance through an increased risk of burnout among medical staff; reduced quality of treatment delivered, either due to burnout, rushing of treatment, reduced facing time, or task shifting resulting in officers performing treatment without adequate training; medical professionals leaving the public sector as a consequence of overworking, acerbating understaffing problems, and hence burden on remaining professionals; and an impact on the satisfaction/experience of care by patients, which may impact their compliance with recommended treatment and subsequent health-seeking behaviour. Failure to capture these effects would therefore equally result in an overestimation of the health gains that can be delivered by the healthcare system. We therefore believe that our current approach provides an initial but balanced first step toward a realistic representation of resource constraints for the Malawian healthcare system, and we postpone an analysis of the above-mentioned factors to a later work.

Finally, an optimal healthcare system should aim to maximise the care delivered without relying on the over-working of its medical workforce, or the reduction of time taken to deliver HSIs, to do so. The adoption of a rigid healthcare system assumption, therefore, is by definition fully consistent with the objective of our optimisation analysis.

## 4.2. Persistence in health-care seeking

Playing an important role in the overall health outcome under a rigid healthcare system is the assumed persistence in health seeking, or how we model the behaviour of patients who have not received treatment on the day they first sought care due to capabilities having been exhausted: are patients likely to persist in seeking care, and if so, with what frequency will they do so? Will they continue to seek care at the same facility, or will they switch to a different provider? How does this behaviour depend on individual characteristics and the medical condition for which they seek treatment? We limited ourselves to considering a single scenario where patients continued to seek care on successive days for a maximum of seven days for all medical conditions. This scenario fails to capture a number of factors: individuals with chronic or long-term conditions may delay seeking treatment again by weeks or months, which may allow for a deterioration of their health in the meantime, potentially leading to a lower success rate in treatment in the long run. In the case of infectious diseases, the delay may lead to an increase in overall incidence of the disease among the population. Additionally, patients are more or less likely to persist in seeking care depending on the severity of their symptoms. If this were to be implemented, we would see capabilities requested by less serious conditions naturally be freed up as a result of patients giving up seeking care in those circumstances, meaning that patients persisting for longer for more serious or chronic conditions would have a higher probability of accessing those resources as a result of their persistence. Unfortunately, no data are currently available to inform these dynamics for each of the conditions captured by the simulation; hence, we postpone an analysis of this assumption to later work.

## 4.3. Enforcing resource constraints based on HCW capabilities

While this analysis, unlike other HBP design approaches, does not enforce a strict budgetary constraint, it does ensure that the health gain provided by the simulated healthcare system is

strictly bound by current expenditures on human resources for health. These, together with healthcare facilities and hospitals, constitute some of the most inelastic resources within the management of the healthcare system, as they require a significant amount of time and investment to train and recruit and must be continuously financially supported once hired. Human resources for health further represent the HSSP area with the highest proportion (68%) of government funding [37]; health worker salaries and benefits, in particular, constitute the largest portion of this cost (70%), and are covered at 90% by the GoM. An analysis that focuses on informing the optimal allocation of this resource is therefore highly relevant to future GoM discussions around obtaining the largest possible health return from its investment. However, a costing analysis of each intervention included in the TLO model is underway, and future expansion of this work plans to include full budgetary constraints.

### 4.4. Practical implementation of prioritisation strategies considered

Our analysis considered a more nuanced formulation of the standard HBP, in which services can be assigned a relative priority as well as being included or excluded altogether. This approach better reflects the type of service delivery that is likely to take place in Malawi, where a strict application of service exclusion is unlikely to be achievable in practice [38, 39]. In line with this, the GoM has been exploring options of moving away from a rigidly defined positive list of prioritised services specific to disease conditions guaranteed for free to the entire population towards different ways of defining packages including implementing user fees for less vulnerable populations and to pivot towards a more patient-centred approach to care prioritisation [18], making the more flexible definition of HBPs adopted in this work more relevant to future policy discussions.

In our analysis, the prioritisation of different services was enforced, for modelling expediency, at the facility level on a daily basis. Furthermore, no realistic cap was imposed on how many high priority services could be delivered by each policy: under the VP policy, for example, we did not accurately reflect the relative allocation of resources between vertical and horizontal programmes currently observed in Malawi today, and hence we may have overestimated how many of these high priority services would actually take place.

While limited, this approach still allowed us to evaluate the possible consequences that concentrating resources on specific services may have on the overall health outcome of the population, while exposing the areas of care which would most benefit from a reallocation of resources away from high-profile ones (such as lower respiratory infections in the case of the VP policy). In reality, the same outcome of relative service delivery could be achieved through a variety of implementations, whether by creating dedicated clinics, boosting training of staff in specific areas, or funding programs to deliberately target vulnerable categories.

### 4.5. Additional points for discussion

In this analysis, we assume a perfect implementation of services (see item iv) in section 1), such that these are in principle always available at the relevant facility level—provided they are provisioned by the prioritisation-policy under consideration, that HCWs required to deliver them are present, and that essential consumables are available. We postpone an analysis of the effect of realistic implementation challenges on health outcomes, which is indeed feasible in the TLO simulation, to a future work. Should the probability of implementation vary across treatment types, we would expect this to have repercussions on our evaluation outcome. Due to a lack of data available to quantify this probability, we unfortunately cannot speculate on the magnitude of this effect.

Another important assumption concerned capabilities being constant over the entire simulated period, while in reality we may reasonably assume that these may be scaled to the growing population over a 20-year period. Because, however, the assumed fixed resources were consistent across different policies, we may expect the limits that this imposes on the health gain achievable to be equal across all policies evaluated. On the other hand, divergent population sizes across different policies considered would result in harsher competition for medical-officers time under policies in which the population is growing more rapidly. We have however found this wouldn't significantly affect our conclusions, as discussed in detail in S5 Appendix.

## 5. Conclusions

Using the TLO simulation, we evaluated a number of resource allocation strategies for Malawi (summarised in section 2.4) which either reflect current preferences in the prioritisation of services among policy-makers and external funding agents (RMNCH, VP, and CMD policies), mirror current analytical tools and approaches to prioritise health interventions (HSSP-III HBP or LCOA), or implement new strategies of prioritisation based on patient characteristics rather than service type (CV policy). In our evaluation, we assumed that prioritisation policies are enforced consistently for a 20-year period (2023–2042). We found that:

- Over the entire simulated 20-year period, the implementation of a number of policies (namely LCOA, RMNCH, CV, and VP) resulted in a significant net health gain—quantified in total DALYs incurred—over a complete lack of prioritisation strategy, with LCOA providing the largest mean possible reduction in overall DALYs incurred ($\sim 8\%$). This suggests that carefully chosen prioritisation strategies may lead to some improvements in health outcomes achievable from existing healthcare system capacities.

- Key to the better performance of the LCOA policy was the more targeted selection of high-priority treatments, and a more nuanced prioritisation strategy (which differentiated between high priority treatments, low priority treatments, and completely excluded treatments), which resulted in resources being more effectively safeguarded for treatments most likely to result in a larger returns in health. For other policies, the high number of high-priority appointments competing for the same resources and a lack of further differentiation at the lower priority level meant that less regulation in resource access could be enforced between them.

- The relative performance of different policies varied significantly during the simulated period, and was shaped by how the relative contribution of key areas of health to the overall health evolved during that time. This implies that our conclusions on the relative performance of the evaluated policies is strongly dependent on the event horizon considered: on a three- to five-year event-horizon basis LCOA, the best performing policy on a 20-year horizon, would actually bring an excess in health burden incurred or no significant advantage in reducing the incurred health burden compared to no prioritisation strategy, respectively. • On the other hand, we found that an approach focused on prioritising vulnerable categories of patients rather than specific services was more resilient to changes in the epidemiological landscape and health challenges faced over time, effectively addressed risk factors, and hence offered consistent improvements over a NP approach year-to-year throughout the simulated period, while resulting in a 5% DALYs reduction overall. This finding strengthens the case for an expansion from more traditional formulations of HBPs, based on the inclusion or exclusion of specific services, towards a more patient-centred approach.

- Lack of consumables availability and poor quality of care played an important part in limiting the performance of some of the policies, suggesting that addressing these limitations could play an important part in increasing the health gain achievable through a prioritisation strategy alone.

- The combination of a patient-centred approach espoused by the CV policy, combined with a more targeted and nuanced prioritisation strategy adopted by the LCOA policy and potential improvements in consumables availability is therefore likely to offer the highest possible return in health in the future.

The evaluation of the policies considered accounted for imperfect clinical practices motivated by country-specific data [26], as well as the probability of consumables stock-outs [28]. Our TLO-based approach, however, further allows us to evaluate how the relative performance of the policies would change were health-seeking and clinical practices to be significantly improved, and hence assess how much the outcome of specific policies is limited not by the design of the policies themselves, but by other factors related to the practical application of those policies. These will be addressed in future work.

We have shown that the TLO simulation provides a unique tool with which to test HBPs designed specifically for Malawi, providing detailed evaluations on the health gains that can be achieved through the relative prioritisation of services given realistic constraints in available human resources for health while addressing limitations in other analytic methods commonly used to weigh evidence for different interventions. Although considerations of equity were not included in the analysis at this stage, they are already included in the TLO model, and will be the focus of later studies. The model is further being expanded to include a full costing analysis for the interventions considered, which will result in a more comprehensive analysis.

## Supporting information

**S1 Appendix. Competition for limited HCWs time: understanding rates of service delivery under resource constraints**
(DOCX)

**S2 Appendix. Mapping of interventions modelled in the TLO simulation to the HSSP-III.**
(PDF)

**S3 Appendix. Motivation and breakdown of individual prioritisation policies considered.**
(DOCX)

**S4 Appendix. Understanding the relative performance of different prioritisation policies in key areas of health.**
(DOCX)

**S5 Appendix. Accounting for diverging population sizes across the evaluated policies.**
(DOCX)

## Acknowledgments

We thank our colleagues in the Government of Malawi Ministry of Health, including the Planning and Policy Development Department and acknowledge the expertise shared by Dr Rose Nyirenda, Dr Mike Chisema, Dr Lazarus Juziwelo, Dr Godfrey Kadewere and Prof Andreas Jahn. We would also like to express our gratitude to the collaborators, attendees and organisers of the Kamuzu University of Health Sciences HEPU Think Tanks which have improved model design, development and analysis immeasurably. We thank Professor Edward Gregg (Royal

College of Surgeons in Ireland, and Imperial College London) and Professor Amelia Crampin (Malawi Epidemiology and Intervention Research Unit, Kamuzu University of Health Sciences, and University of Glasgow) for useful discussions.

## Author Contributions

**Conceptualization:** Margherita Molaro, Timothy B. Hallett.

**Data curation:** Margherita Molaro.

**Formal analysis:** Margherita Molaro.

**Funding acquisition:** Timothy B. Hallett.

**Investigation:** Margherita Molaro.

**Methodology:** Margherita Molaro, Sakshi Mohan, Bingling She, Martin Chalkley, Andrew N. Phillips, Tara D. Mangal, Timothy B. Hallett.

**Software:** Margherita Molaro, Joseph H. Collins, Matthew M. Graham, Eva Janoušková, Ines Li Lin, Emmanuel Mnjowe, Asif U. Tamuri, Tara D. Mangal.

**Supervision:** Timothy B. Hallett.

**Visualization:** Margherita Molaro.

**Writing – original draft:** Margherita Molaro.

**Writing – review & editing:** Margherita Molaro, Sakshi Mohan, Bingling She, Martin Chalkley, Tim Colbourn, Joseph H. Collins, Emilia Connolly, Matthew M. Graham, Eva Janoušková, Ines Li Lin, Gerald Manthalu, Emmanuel Mnjowe, Dominic Nkhoma, Pakwanja D. Twea, Andrew N. Phillips, Paul Revill, Asif U. Tamuri, Joseph Mfutso-Bengo, Tara D. Mangal, Timothy B. Hallett.

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
