## [Decision Letter · Decision Letter 0]

8 Apr 2024

Dear Dr Molaro,

Thank you very much for submitting your manuscript "A new approach to Health Benefits Package design: an application of the Thanzi La Onse model in Malawi" for consideration at PLOS Computational Biology.

As with all papers reviewed by the journal, your manuscript was reviewed by members of the editorial board and by several independent reviewers. In light of the reviews (below this email), we would like to invite the resubmission of a significantly-revised version that takes into account the reviewers' comments.

We cannot make any decision about publication until we have seen the revised manuscript and your response to the reviewers' comments. Your revised manuscript is also likely to be sent to reviewers for further evaluation.

Sincerely,

Samuel V. Scarpino

Academic Editor

PLOS Computational Biology

Virginia Pitzer

Section Editor

PLOS Computational Biology

Editorial Comments:

I agree with the reviewers that this is a potentially important contribution related to the effectiveness of health benefit packages. However, I also agree that more work is needed to support a number of the conclusions described in the abstract. In particular, providing more specifics around which methods are considered insufficient and which methodological comparisons are performed in the paper. Additionally, the reviewers raised concerns about why results on burden of disease differed from those published by the Institute for Health Metrics and Evaluation (IHME). Explaining this discrepancy will be crucial for future evaluations; however, I want to stress that while IHME data are often considered "gold standard" the authors may in fact be correct about the true trends. I also agree that more work is needed to compare to existing methods and cite existing/similar approaches.

Reviewer's Responses to Questions

**Comments to the Authors:**

Reviewer #1: see word document, uploaded separately

Reviewer #2: My review comments have been uploaded as an attachment

**Have the authors made all data and (if applicable) computational code underlying the findings in their manuscript fully available?**

Reviewer #1: Yes

Reviewer #2: **No: **The authors indicated that they will make the data and code available upon acceptance of the journal article.

PLOS authors have the option to publish the peer review history of their article (what does this mean?). If published, this will include your full peer review and any attached files.

Reviewer #1: No

Reviewer #2: No
---

## [Decision Letter · Decision Letter 1]

5 Aug 2024

Dear Dr Molaro,

Thank you very much for submitting your manuscript "A new approach to Health Benefits Package design: an application of the Thanzi La Onse model in Malawi" for consideration at PLOS Computational Biology. As with all papers reviewed by the journal, your manuscript was reviewed by members of the editorial board and by several independent reviewers. The reviewers appreciated the attention to an important topic. Based on the reviews, we are likely to accept this manuscript for publication, providing that you modify the manuscript according to the review recommendations.

I agree that a short paragraph on whether this approach is relevant for reinfection risk is worth the revision (along with the items identified by the reviewer in their attached document)

Sincerely,

Samuel V. Scarpino

Academic Editor

PLOS Computational Biology

Virginia Pitzer

Section Editor

PLOS Computational Biology

I agree that a short paragraph on whether this approach is relevant for reinfection risk is worth the revision (along with the items identified by the reviewer in their attached document)

Reviewer's Responses to Questions

**Comments to the Authors:**

Reviewer #1: I applaud the authors' extensive response to all the issues raised by myself and the other reviewer in their rebuttal. The article has much improved and I have no further questions or suggestions. It is a welcome addition to the literature and I look forward to seeing it published.

Reviewer #3: Excellent paper. I have noted a few grammatical suggestions. I think you should add a paragraph or two about whether this approach could be used when reinfections occur -- e.g., schistosomiasis

**Have the authors made all data and (if applicable) computational code underlying the findings in their manuscript fully available?**

Reviewer #1: Yes

Reviewer #3: Yes

PLOS authors have the option to publish the peer review history of their article (what does this mean?). If published, this will include your full peer review and any attached files.

Reviewer #1: **Yes: **Leon Bijlmakers

Reviewer #3: **Yes: **Jay Richard Stauffer, Jr.

Figure Files:

Data Requirements:

Reproducibility:

References:

---

## [Editor Report · Decision Letter 2]

5 Sep 2024

Dear Dr Molaro,

We are pleased to inform you that your manuscript 'A new approach to Health Benefits Package design: an application of the Thanzi La Onse model in Malawi' has been provisionally accepted for publication in PLOS Computational Biology.

Best regards,

Samuel V. Scarpino

Academic Editor

PLOS Computational Biology

Virginia Pitzer

Section Editor

PLOS Computational Biology

---

## [Editor Report · Acceptance letter]

23 Sep 2024

PCOMPBIOL-D-24-00243R2 

A new approach to Health Benefits Package design: an application of the Thanzi La Onse model in Malawi

Dear Dr Molaro,

I am pleased to inform you that your manuscript has been formally accepted for publication in PLOS Computational Biology. Your manuscript is now with our production department and you will be notified of the publication date in due course.

With kind regards,

Anita Estes
